# Rearrangement Planning for General Part Assembly

**Yulong Li**[1]      **Andy Zeng**[2]      **Shuran Song**[1]
[1]Columbia University      [2]Google Deepmind
https://general-part-assembly.github.io/

**Abstract:** Most successes in autonomous robotic assembly have been restricted to single target or category. We propose to investigate general part assembly, the task of creating novel target assemblies with unseen part shapes. As a fundamental step to a general part assembly system, we tackle the task of determining the precise poses of the parts in the target assembly, which we term "rearrangement planning". We present General Part Assembly Transformer (GPAT), a transformer-based model architecture that accurately predicts part poses by inferring how each part shape corresponds to the target shape. Our experiments on both 3D CAD models and real-world scans demonstrate GPAT's generalization abilities to novel and diverse target and part shapes.

## 1   Introduction

The ability to assemble new objects is a hallmark of visuo-spatial reasoning. With the mental image of a novel target shape, one can arrange possibly unseen parts at hand to create a resembling assembly, either building an alien spaceship with lego blocks or a rain shelter with stones. Building autonomous robotic systems that exhibit these capabilities may give rise to wide range of robotics applications from autonomously assembling new objects in a manufacturing plant to building shelter in disaster response scenarios.

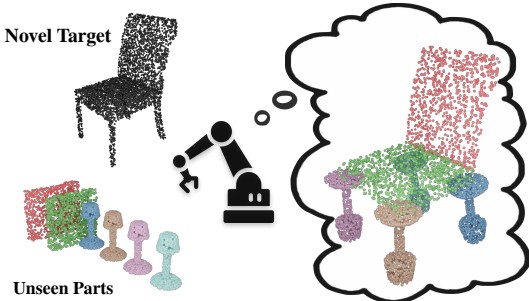

Figure 1: **General Part Assembly**. We seek to build autonomous robotic systems that can assemble a novel target with previously unseen parts. The visualizations are actual inputs and prediction of our model.

Despite the interest and progress in part assembly, existing methods tend to focus on *specialized part assembly* consisting of fixed targets [1, 2] or seen categories [3, 4]. We propose to instead investigate the task of *general part assembly*, which takes in as inputs both a target shape and a variable set of part shapes to build an assembly resembling the target. Instead of restricting to fixed objects or categories, we require the robotic system to generalize to novel target shapes without additional annotation or supervision. Moreover, the available parts are not guaranteed to be carefully manufactured, and the robotic system has to use parts of slightly differing shapes, the *non-exact parts*, e.g., building a table with rectangular blocks given a round table as the target.

The task of *general part assembly* is an extension of *specialized part assembly* that focus on fixed targets or categories. For fixed-target assembly, the target shape information is implicitly provided to the agent. A general part assembly agent can also solve category-level part assembly by taking in as input a single instance of the category, while a typical learning method is trained on a large number of instances from the category [3, 4]. For example, given a single table instance, a general part assembly agent can assemble tables with either rectangular tabletop or round tabletop.

In this work, we focus on the initial perception and planning module for general part assembly, which outputs the precise poses of the parts in the target assembly. Our key insight is to formulate this module as a goal-conditioned shape rearrangement problem, whereby the target can be viewed as a desired 3D shape layout. Consequently, we term this module "rearrangement planing", which

7th Conference on Robot Learning (CoRL 2023), Atlanta, USA.

aligns with established definitions previously proposed by the Task and Motion Planning (TAMP) community [5]. With this insight, the module factorizes into two steps: predict a segmentation of the target, where each segment corresponds to a part, and infer the pose of each part with pose estimation. To predict accurate segmentation of the target, a key challenge is to deal with the ambiguities in the target shape due to geometrically equivalent parts (e.g., legs of a table). To infer accurate segmentations, we propose General Part Assembly Transformer (GPAT), a transformer-based model architecture, that processes input shapes in a fine-to-coarse manner, thereby ensuring consistent segmentation results.

To train and evaluate our model, we build a benchmark based on PartNet [6], a large-scale dataset of 3D objects with part information. We programmatically generate primitive part shapes as non-exact parts. We demonstrate that GPAT generalizes well to entirely new target structures at random orientations and novel parts that are non-exact matches on both synthetic and real-world data.

In summary, our primary contributions are three-fold:

- We propose the task of general part assembly to study the ability of building novel targets with unseen parts and create a benchmark based on PartNet [6].
- We tackle the planning problem for general part assembly as a goal-conditioned shape rearrangement problem – treating part assembly as an "open-vocabulary" (i.e., vocabulary of parts) target object segmentation task.
- We introduce General Part Assembly Transformer (GPAT) for assembly planning, which can be trained to generalize to novel and diverse target and part shapes.

We believe that GPAT is an exciting step for general part assembly – we discuss both its capabilities and limitations in the report.

## 2   Related Work

**Specialized Part Assembly.** A number of learning-based methods have been proposed for part assembly, but they usually have limited generalization abilities, so we refer to them as specialized part assembly. Reinforcement learning (RL) has success in building part assembly for fixed targets [1, 2, 7, 8] or seen categories [9]. They require costly trial-and-error in real-world or physics-based environments to extend to novel targets, and often require low-level state information during training and testing. Another line of work directly works with visual perception and learns shape correspondences, which has success in tasks like kit assembly [10, 11] and shape mating [12], but they have not tackled part assembly which involves more complex and diverse targets and parts. Previously, part assembly with category-level generalization is tackled with models based on graph neural network (GNN) backbones [3, 4, 13]. Notably, Li et al. [3] and our method shares the same high-level idea of segmenting the target shape, even though their targets are represented as images. Additionally, Funk et al. [14] proposed a full robotic system based on GNN and RL to assemble arbitrary target blueprints with rectangular blocks. In this work, we propose to tackle general part assembly with novel and semantically grounded target and part shapes.

**Part Assembly with Object Models.** Physics-based part assembly assumes precise models, and the goal poses of the parts are explicitly given or implicitly derived. However, these requirements hinder quick generalization to novel targets and parts. We directly work with visual perception, the 3D point clouds, to predict the precise poses of the parts. With our prediction, one may apply physics-based assembly sequence planning [15, 16, 17, 18] and path planning [19, 20, 21, 22] to obtain a complete assembly plan.

**Point cloud Registration.** Point cloud registration estimates the transformation matrix between two point clouds from different views of the same 3D scene. It is traditionally solved by optimization-based methods and recently by learning-based methods [23]. If we represent target and part shapes as point clouds, general part assembly is akin to point cloud registration, but it crucially demands optimizing the part poses concurrently. As demonstrated in Sec. 4, basic alterations to point cloud registration methods can't directly address general part assembly.

# 3 Approach

Given a target point cloud $\mathcal{T}$ and part point clouds $\{\mathcal{P}_i\}_{i=1}^{N}$ as inputs, where $N$ denotes the number of input parts and varies for different shapes, the goal of our task is to predict a 6-DoF part pose $q_i \in SE(3)$ for each input part $\mathcal{P}_i$, forming a final part assembly, $\mathcal{P} = \bigcup_{i=1}^{N} q_i(\mathcal{P}_i)$, where $q_i(\mathcal{P}_i)$ denotes the transformed part point cloud. To tackle this problem, we propose to solve part assembly in two steps: target segmentation (Sec. 3.1) – which utilizes General Part Assembly Transformer (Sec. 3.2) to decompose the target into disjoint segments, each representing a transformed part – and pose estimation (Sec. 3.3) to obtain the final part poses.

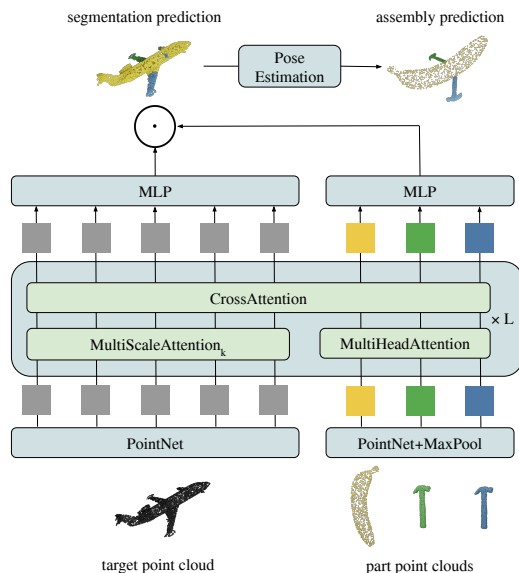

Figure 2: **Method Overview.**

## 3.1 Part Assembly by Target Segmentation

Given a target point cloud $\mathcal{T}$ and part point clouds $\{\mathcal{P}_i\}_{i=1}^{N}$ as inputs, we want to segment the target point cloud such that each segment corresponds to a part. From the segmentation, we can infer the part pose with pose estimation, which is a transformation from the part to the corresponding segment.

Formally, we want to predict a set of disjoint segments $\{\mathcal{T}_i\}_{i=1}^{N}$ such that $\bigcup_{i=1}^{N} \mathcal{T}_i = \mathcal{T}$ and $\mathcal{T}_i \cap \mathcal{T}_j = \emptyset$ for $i \neq j$. Further $\{\mathcal{T}_i, \mathcal{P}_i\}_{i=1}^{N}$ represents a bipartite matching between the segments and the parts. Note that $\mathcal{T}_i = \emptyset$ may be empty, suggesting that the part $\mathcal{P}_i$ is not used in the assembly. The goal of target segmentation is to maximize the geometric resemblance for each pair with non-empty $\mathcal{T}_i$, as defined by the following minimization problem,

$$\{\mathcal{T}_i\}_{i=1}^{N} = \arg\min_{\mathcal{T}_1, \ldots, \mathcal{T}_N} \sum_{i=1, \mathcal{T}_i \neq \emptyset}^{N} \min_{q_i} dist(\mathcal{T}_i, q_i(\mathcal{P}_i))$$

where $dist$ is some distance metric for point clouds (e.g., chamfer distance). This may appear to be a detour since the goal of the task is the transformation $q_i$, which is implicitly optimized over. Nevertheless, a model can approximate $\min_{q_i} dist(\mathcal{T}_i, q_i(\mathcal{P}_i))$ by learning rotationally invariant representations of point clouds to avoid optimization over part poses. Finally, we can infer a part pose $q_i$ by minimizing $dist(\mathcal{T}_i, q_i(\mathcal{P}_i))$.

## 3.2 General Part Assembly Transformer (GPAT)

The input to General Part Assembly transformer is a target point cloud and a set of part point clouds, and therefore model backbones designed for a single point cloud is insufficient for the task. GPAT uses PointNet [24] to extract initial features for the point clouds, and then leverages a transformer-based architecture [25] to jointly optimize over the target shape and the part shapes. To predict accurate target segmentation, a key challenge is the ambiguities in the target shape (e.g., the four legs of a chair are interchangeable). As a result, a target shape often admits multiple ground-truth segmentations, and a fine-grained and consistent segmentation of the target is required for successful assembly. In light of this, we design the GPAT layer to fully exploit the spatial structure of the target point cloud from a fine-to-coarse manner, which is inspired by the increasing receptive field of convolutional neural networks [26] and progress in hierarchical feature learning for point clouds [27, 28].

More formally, let the hidden dimension for features be $h$. For a query feature $\mathbf{q} \in \mathbb{R}^h$ and a set of $k$ key features $\mathbf{K} \in \mathbb{R}^{k \times h}$, we denote the dot-product attention operator as

$$\text{Attention}(\mathbf{q}, \mathbf{K}) = W_v(\mathbf{K})^T \text{softmax}\left(\frac{W_k(\mathbf{K}) \cdot W_q(\mathbf{q})}{\sqrt{h}}\right)$$

where $W_q, W_k, W_v$ are MLPs.

Given a target $\mathcal{T}$ and a set of parts $\{\mathcal{P}_i\}_{i=1}^N$, GPAT uses Pointnet to extract an initial target point feature $\mathbf{v}_t^0$ for each point $X_t \in \mathcal{T}$, and an initial part feature $\mathbf{u}_i^0$ for each part $\mathcal{P}_i$ (with max pooling). Then the features pass through $L$ GPAT layers. At the $(n+1)$-th GPAT layer, we have a target point feature $\mathbf{v}_t^n$ for each point $X_t \in \mathcal{T}$ and part feature $\mathbf{u}_i^n$ for each part $\mathcal{P}_i$. The features are updated in three steps. The first step is multi-scale attention which is parameterized by a positive integer $k$ and denoted by $\text{MultiScaleAttention}_k$ in Fig. 2. It updates the target point features as follows:

$$\overline{\mathbf{v}_t^n} = \text{Attention}(\mathbf{v}_t^n, \mathcal{N}_k(\mathbf{v}_t^n))$$

where $\mathcal{N}_k(\mathbf{v}_t^n)$ is the features of the $k$ nearest neighbors of the target point $X_t$. GPAT gradually increases $k$ to let each point receive global information of the point cloud. The second step is the multi-head attention [25], which updates the part features. In the final step, GPAT applies attention updates between the target point features and part point cloud features, denoted as CrossAttention in Fig. 2:

$$\mathbf{v}_t^{n+1} = \text{Attention}(\overline{\mathbf{v}_t^n}, \overline{\mathbf{U}^n}) \quad \mathbf{u}_i^{n+1} = \text{Attention}(\overline{\mathbf{u}_i^n}, \overline{\mathbf{V}^n})$$

where $\overline{\mathbf{V}^n}$ denotes all target point features and $\overline{\mathbf{U}^n}$ denotes all part point cloud features. Finally, GPAT models how likely a point $X_t$ is matched with an input part $\mathcal{P}_i$ as

$$\mathbb{P}[X_t \in \mathcal{T}_i] = \frac{W_T(\mathbf{v}_t^L) \cdot W_P(\mathbf{u}_i^L)}{\sum_{j=1}^N W_T(\mathbf{v}_t^L) \cdot W_P(\mathbf{u}_j^L)}$$

where $W_T$ and $W_P$ are MLP projections.

**Data Augmentation.** In order to generalize to targets at random poses, we augment the dataset by randomly rotating the target point cloud but keep the order of the points. Thus the ground truth segmentation label is unchanged, which encourages the model to obtain rotationally invariant feature representations for the target. We always preprocess the input parts so that their principle axes are aligned with world axes. Further, to generalize to unseen categories and non-exact parts, the dataset needs to comprise diverse shapes. We programmatically generate rectangular and spherical primitive shapes of various sizes. For each data sample with exact parts, we construct a new sample with each exact part replaced with the primitive of the most similar sizes. GPAT is trained with both data samples of exact and non-exact parts. Assemblies with non-exact parts can be found in Fig. 3.

**Training and Loss.** To supervise GPAT, we use the per-point cross entropy loss between the predicted distribution over all parts and the ground truth label. Denote the ground truth segmentation by $\{\mathcal{T}_i^{gt}\}_{i=1}^N$, and the loss function is

$$\mathcal{L} = -\sum_{t=1}^{|\mathcal{T}|} \log \mathbb{P}[X_t \in \mathcal{T}_i^{gt}]$$

For part assembly, there are often multiple ground-truth labels due to geometric equivalence between parts (e.g., legs of a chair). We enumerate all permutations of labels corresponding to the geometrically equivalent parts and adopt the lowest possible cost.

## 3.3 Predicting Part Assembly with Segmentation

Given a set of parts $\{\mathcal{P}_i\}_{i=1}^N$ and a segmentation of the target $\{\mathcal{T}_i\}_{i=1}^N$, we can find the 6-DoF part pose $q_i \in SE(3)$ for each part with pose estimation. Since the parts in our task are not necessarily exact, we use oriented-bounding boxes to estimate part poses which is simple and robust. For each non-empty $\mathcal{T}_i$, we use principle component analysis to find the oriented bounding boxes of $\mathcal{P}_i$ and $\mathcal{T}_i$ to solve for $q_i$. In practice, we improve the bounding box predictions by filtering the outliers (points that are at least one standard deviation away from the center) in $\mathcal{T}_i$.

| | Unseen Instance | | | | | | | | | | | | Unseen Category | | | | | | | | | | | |
|---|---|---|---|---|---|---|---|---|---|---|---|---|---|---|---|---|---|---|---|---|---|---|---|---|
| | Canonical Pose | | | | | | Random Pose | | | | | | Canonical Pose | | | | | | Random Pose | | | | | |
| | Precise Part | | | Imprecise Part | | | Precise Part | | | Imprecise Part | | | Precise Part | | | Imprecise Part | | | Precise Part | | | Imprecise Part | | |
| | CD | PA | SR | CD | PA | SR | CD | PA | SR | CD | PA | SR | CD | PA | SR | CD | PA | SR | CD | PA | SR | CD | PA | SR |
| Opt | 7.9 | 18.3 | 2.4 | 10.0 | 16.0 | 0.9 | **6.3** | 22.9 | 3.7 | **7.7** | 21.0 | 2.8 | **5.1** | 23.1 | 5.7 | **5.4** | 21.3 | 4.7 | **4.1** | 31.3 | 6.7 | **5.0** | 28.4 | 4.2 |
| Go-ICP | 72.9 | 4.2 | 0.1 | 67.9 | 3.9 | 0.0 | 72.3 | 4.4 | 0.1 | 66.2 | 3.9 | 0.0 | 49.4 | 2.0 | 0.0 | 42.6 | 2.6 | 0.0 | 45.7 | 2.2 | 0.0 | 39.3 | 2.6 | 0.0 |
| GeoTF | 54.3 | 14.5 | 4.1 | 63.4 | 9.5 | 1.6 | 53.7 | 14.9 | 4.2 | 62.6 | 10.1 | 1.9 | 57.3 | 2.8 | 0.1 | 53.8 | 2.3 | 0.0 | 57.4 | 2.9 | 0.1 | 53.9 | 2.9 | 0.2 |
| NSM | 89.8 | 1.3 | 0.0 | 86.3 | 0.9 | 0.0 | 87.1 | 1.4 | 0.0 | 83.1 | 0.9 | 0.0 | 58.0 | 0.7 | 0.0 | 52.0 | 1.1 | 0.0 | 56.4 | 1.1 | 0.0 | 49.7 | 1.4 | 0.0 |
| DGL | 21.5 | 45.4 | 10.9 | 18.0 | 51.6 | 14.4 | 86.7 | 1.1 | 0.0 | 75.5 | 1.6 | 0.2 | 27.2 | 13.7 | 1.1 | 22.1 | 18.2 | 0.7 | 48.9 | 1.6 | 0.0 | 45.0 | 1.9 | 0.0 |
| DGL-aug | 53.4 | 6.7 | 0.6 | 44.3 | 8.1 | 0.5 | 52.3 | 7.0 | 0.6 | 44.0 | 8.7 | 0.2 | 31.3 | 7.3 | 0.1 | 26.4 | 9.3 | 0.3 | 28.4 | 8.5 | 0.2 | 23.9 | 11.1 | 0.3 |
| Reg | 33.6 | 3.1 | 0.3 | 33.5 | 3.2 | 0.5 | 25.6 | 5.0 | 0.2 | 25.7 | 5.8 | 0.5 | 34.5 | 1.9 | 0.0 | 31.4 | 3.3 | 0.2 | 19.0 | 5.4 | 0.1 | 18.7 | 5.0 | 0.2 |
| TF | 11.4 | 47.9 | 16.8 | 11.5 | 45.4 | 14.4 | 9.1 | 57.8 | 21.5 | 9.7 | 54.7 | 18.8 | 13.5 | 31.8 | 5.1 | 12.3 | 33.3 | 4.9 | 14.1 | 28.0 | 4.2 | 12.2 | 29.9 | 3.7 |
| Ours | **7.6** | **61.6** | **23.2** | **7.2** | **64.8** | **26.0** | 7.8 | **60.8** | **21.7** | 7.8 | **64.3** | **26.0** | 7.1 | **53.4** | **20.1** | 6.6 | **56.3** | **21.7** | 7.6 | **52.2** | **18.8** | 6.9 | **55.6** | **19.8** |

Table 1: **Quantitative Results and Comparisons.** We adopt three metrics: chamfer distance (CD) measured in ‰, part accuracy (PA) measured in %, and success rate (SR) measured in %.

# 4 Evaluation

**Tasks.** For both training and quantitative evaluation, we use PartNet [6], a large-scale dataset of 3D objects with fine-grained and instance-level 3D part information. We use chairs, lamps, and faucets for training and hold out tables and displays as novel categories. We deal with the most fine-grained level of PartNet segmentation, and adopt the default train/test split of the PartNet, which contains 2463 instances of chairs, 1553 instances of lamps, and 510 instances of faucets. We categorize generalization scenarios across three dimensions.

- **Novel target instances or categories:** We evaluate on the unseen instances of chairs, lamps, and faucets, and two novel categories: tables and displays.
- **Random target poses:** We evaluate on targets at either canonical orientation (as defined in the dataset) or a random orientation uniformly sampled from $SO(3)$.
- **Non-exact parts:** Besides exact parts from the dataset, we programatically generated rectangular and spherical blocks as non-exact parts. Sample instances can be found in Fig. 3.

**Metrics.** For all the tasks, we measure the quality of the predicted assembly with three metrics: chamfer distance, part accuracy, and assembly success rate.

- **Chamfer distance (CD):** Given two point clouds $\mathcal{A}, \mathcal{B}$, the chamfer distance between $\mathcal{A}$ and $\mathcal{B}$ is

$$CD(\mathcal{A}, \mathcal{B}) = \sum_{x \in \mathcal{A}} \min_{y \in \mathcal{B}} ||x - y||_2^2 + \sum_{y \in \mathcal{B}} \min_{x \in \mathcal{A}} ||x - y||_2^2$$

  We use $CD(\mathcal{T}, \mathcal{P})$ as a metric, abbreviated as $CD$, where $\mathcal{T}$ is the target point cloud, and $\mathcal{P} = \bigcup_{i=1}^{N} q_i(\mathcal{P}_i)$ where $q_i$ is the predicted pose for the $i$-th part.[1]

- **Part accuracy (PA):** Adopted from the previous work [4], part accuracy is defined as,

$$\frac{1}{N} \sum_{i=1}^{N} \mathbb{1}\left(CD(q_i^{GT}(\mathcal{P}_i), q_i(\mathcal{P}_i)) < \tau_p\right)$$

  where $q_i^{GT}$ is the ground truth pose of the $i$-th part, $\tau_p = 0.01$. This metric indicates the percentage of the predicted parts that match the GT part up a certain threshold measured in chamfer distance. Due to possible geometric equivalence between parts (e.g., the legs of a chair), we enumerate all possible labels of geometrically equivalent parts to obtain different GT poses and take the highest accuracy value.

- **Assembly Success Rate (SR):** A predicted assembly is considered successful if its part accuracy (PA) is equal to 1. We report the percentage of successful predictions out of all data samples as the assembly success rate (SR).

**Algorithm comparisons.** Since general part assembly is a novel task, there are no previous methods specifically solving the task. We adapt methods for point cloud registration and specialized part assembly and compare with variants of our method for ablation studies.

---

[1]Note that in the previous work [4], 'shape chamfer distance' is defined differently with $\mathcal{T} = \bigcup_{i=1}^{N} q_i^{GT}(\mathcal{P}_i)$. In our tasks, the target is not a union of the given parts, so the values according to our metric are usually larger.

- **Opt:** Covariant matrix adaptation evolution strategy (CMA-ES) [29] is used to optimize the poses of each part by minimizing the chamfer distance CD as defined above.
- **Go-ICP**: We greedily match each part point cloud to the target point cloud using Go-ICP [30].
- **GeoTF**: Geometric Transformer (GeoTF) [31] is one of the SoTA methods for point cloud registration. We modify the algorithm to simultaneously optimize for all part poses.
- **NSM**: Neural Shape Mating (NSM) [12] uses a transformer-based model to solve pairwise 3D geometric shape mating such as reconstruct two broken pieces of an object. We modify their algorithm to match each part to the target and simultaneously optimize for all part poses.
- **DGL**: Dynamic Graph Learning (DGL) [4] tackles category-level part assembly by leveraging an iterative graph neural network backbone to regress part poses. Designed for category-level generalization, DGL does not take in the target shape as an input. To adapt to our task, we include the target encoding as a node into the graph neural network framework. DGL is trained only with targets at canonical poses following the previous work. DGL-aug uses the same training dataset as our model, with augmentation of targets at random poses.
- **Reg:** Instead of predicting a segmentation of the target for the subsequent pose estimation, we replace the final dot-product segmentation layer of our model with MLPs to directly regress a 6DoF pose for each part. We trained the modified model with the supervision of GT poses.
- **TF:** As an ablation, we replace each GPAT layer with a vanilla transformer layer [25].

## 5  Experimental Results

Tab. 1 and Fig. 3 summarizes the main quantitative and qualitative results, and the following sections provide detailed discussions. Please refer to the supplementary materials for more results.

**Part assembly by target segmentation is more generalizable.** The optimization baseline (Opt) achieves the lowest chamfer distance (CD) in some scenarios, but its part accuracy (PA) and success rate (SR) are significantly lower. Directly optimizing the part poses often result in predictions at local minima where the predicted assembly matches the contour of the target, but the assembly makes no semantic sense. (Please refer to supplementary materials for an example.)

As a classical optimization-based algorithm, Go-ICP tries to match individual part to the entire target which fails with no surprises. As a learning-based method, GeoTF achieves better performances for seen categories, but fails nonetheless for unseen categories. Since the parts are non-exact, it is challenging for point cloud registration methods to find suitable correspondences either in spacial coordinates or a learned feature space.

GPAT also outperforms regression-based models (DGL, DGL-aug, NSM, Reg) across all the tasks, especially at scenarios that requires more generalization (Tab. 1 and Fig. 3). For the most challenging scenario, regression-based models achieve less than 1% success rate, while GPAT has 19.8% success rate which is attained for all the test scenarios. To directly regress poses, a model needs to learn rotationally equivariant features for the target shape. However, given non-exact parts, unseen categories, and targets at random poses, we show that the regression-based models fail to capture the distribution of poses. They either overfit certain canonical poses and assembly structures or fail to learn. In contrast, GPAT, a segmentation-based model, is trained to learn rotationally invariant representations of the shapes, and thus it experiences minor performance drops facing generalization scenarios. Further, training with diverse shapes makes the representations generalizable and applicable to new categories.

**GPAT is robust against targets with ambiguities.** Compared to the alternative segmentation-based model using vanilla transformer, GPAT achieves better results for all test scenarios, especially for unseen target categories (Tab. 1 and Fig. 3). We compare the segmentation accuracy and show the results in Tab. 2 and visualize a typical failure case of TF in Fig. 5. When facing inputs with multi-model ground truths (e.g., a chair with identical legs), TF is unable to produce consistent segmentation, which hinders successful assembly prediction. With GPAT layers, we fully leverage the point cloud structure and process the point cloud in a fine-to-coarse manner, thereby achieving local consistency of the segmentation predictions.

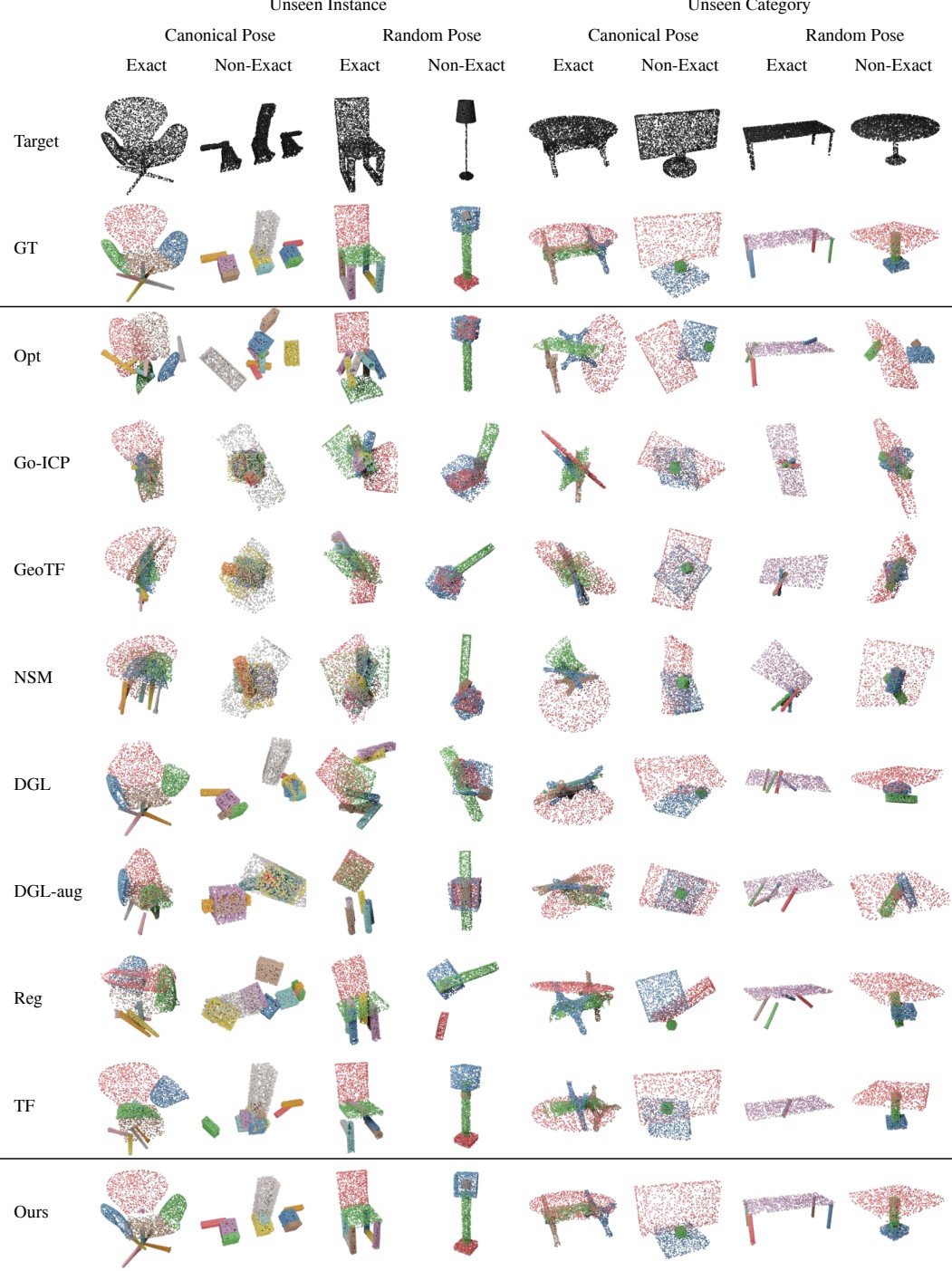

Figure 3: **Assembly Results and Comparisons**. For targets at random poses, targets and predictions are transformed to be *visualized at canonical poses* for better understanding. Please see Fig. 4 for more results of randomly oriented targets. The optimization-based approach (Opt and Go-ICP) tends to stuck at local minima. The learning-based alternatives (GeoTF, NSM, DGL, Reg) overfit the training scenarios and fail to learn rotationally equivariant features for target shapes that are necessary for accurate pose inference. The alternative segmentation-based model that uses the vanilla transformer (TF) fails to produce consistent segmentations for targets with geometrically equivalent parts (see Fig. 5 for a detailed example). With the multi-scale attention layer, GPAT fully leverages the spatial structure of the target point clouds to produce consistent segmentations and accurate assemblies.

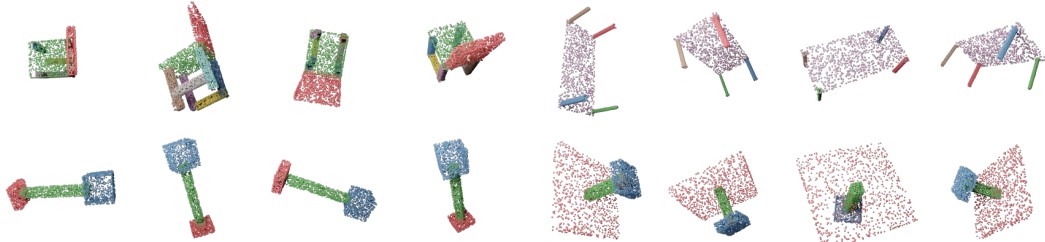

Figure 4: **Assembly Results for Targets at Random Orientations.** We show more results for the same targets in Fig. 3 but at random orientations. GPAT is robust against the orientation of the target shape. More results can be found in the supplementary materials.

**GPAT solves generalizes well to real-world data.** We use the real-world scans from redwood dataset [32] as targets and part point clouds from PartNet. As seen in Fig 6, our method produces diverse assemblies that resemble the target. This result also illustrates how GPAT can be used to assemble different sets of parts given a single target shape from the category.

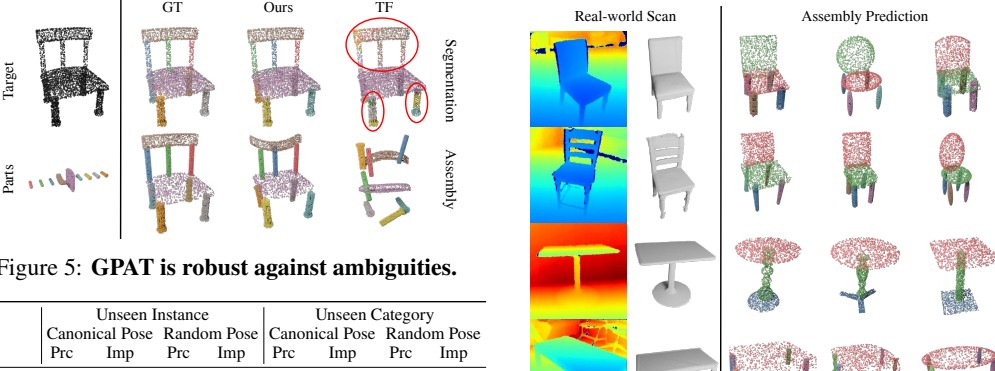

Figure 5: **GPAT is robust against ambiguities.**

Figure 6: **Results on Real-world Data**

|  | Unseen Instance | | | | Unseen Category | | | |
|---|---|---|---|---|---|---|---|---|
|  | Canonical Pose | | Random Pose | | Canonical Pose | | Random Pose | |
|  | Prc | Imp | Prc | Imp | Prc | Imp | Prc | Imp |
| TF | 70.0 | 62.2 | 76.5 | 69.2 | 63.7 | 62.4 | 62.9 | 60.9 |
| Ours | **76.7** | **70.9** | **76.6** | **71.1** | **69.5** | **69.3** | **69.6** | **69.5** |

Table 2: **Segmentation Accuracy (%)**

**Failure mode analysis.** GPAT is not without limitations, and Fig. 7 shows some typical failure cases. First, GPAT tends to give incorrect segmentation predictions if some parts are hidden inside a larger part (e.g., the light bulbs in a lamp) or the parts are less separable (e.g., overlapping parts of a microwave). To solve with these issues, it is possible to introduce additional information like colors and normals of the point clouds as inputs. Additionally, oriented bounding box can be insufficient as a pose estimation method for some parts. To tackle this problem, a learning-based pose estimation module can potentially replace the bounding box procedure.

# 6 Conclusion

In this work, we formulate the task of general part assembly, which focuses on building novel target assemblies with diverse and unseen parts. To plan for a general part assembly task, we propose General Part Assembly Transformer (GPAT) and factorizes the task into target segmentation and pose estimation. Our experiments show that GPAT performs well under all the generalization scenarios. By integrating with an assembly sequence and path planning algorithm, we believe that GPAT has great potential in building vision-based general robotic assembly systems.

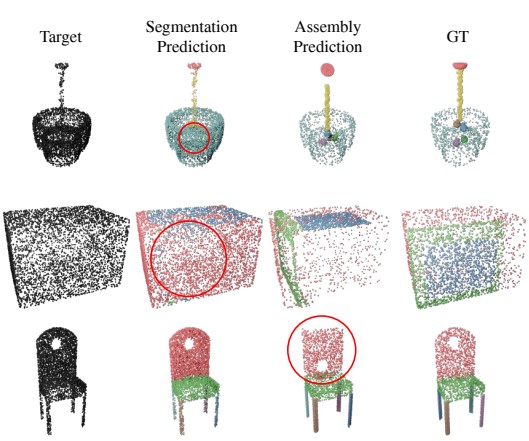

Figure 7: **Typical Failure Cases.**

# 7 Acknowledgement

We would like to thank Huy Ha, Zhenjia Xu, Cheng Chi, and Zeyi Liu for their helpful feedback and fruitful discussions. This work was supported in part by NSF Award #2037101, #2143601, and #2132519. The views and conclusions contained herein are those of the authors and should not be interpreted as necessarily representing the official policies, either expressed or implied, of the sponsors.

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

# 8 Appendix

## 8.1 Additional Results and Analysis

**Additional Visualization.** Fig. 13 and Fig. 14 show additional results on simulated and real-world data, respectively.

**Quantitative Results for Categories.** Table 3 shows detailed quantitative evaluation for unseen instances for seen categories (Chair, Lamp, Faucet) and unseen categories (Table, Display).

Table 3: Quantitative results of our algorithm on different categories.

|  | Canonical Pose | | | | | | Random Pose | | | | | |
|  | Precise Part | | | Imprecise Part | | | Precise Part | | | Imprecise Part | | |
|  | CD | PA | SR | CD | PA | SR | CD | PA | SR | CD | PA | SR |
|---|---|---|---|---|---|---|---|---|---|---|---|---|
| Chair | 7.7 | 57.7 | 19.3 | 7.3 | 64.4 | 25.1 | 7.6 | 58.4 | 19.3 | 8.3 | 63.0 | 24.0 |
| Lamp | 7.6 | 66.9 | 29.1 | 5.9 | 72.1 | 40.9 | 8.1 | 64.2 | 26.2 | 6.0 | 74.4 | 45.5 |
| Faucet | 7.3 | 65.6 | 24.4 | 6.5 | 63.1 | 20.7 | 7.4 | 63.6 | 20.6 | 6.5 | 62.3 | 22.4 |
| Table | 7.5 | 52.0 | 20.6 | 7.0 | 55.1 | 21.5 | 8.1 | 50.8 | 17.8 | 6.9 | 55.7 | 22.2 |
| Display | 4.2 | 59.2 | 23.2 | 4.7 | 59.3 | 20.2 | 4.4 | 60.8 | 23.8 | 4.9 | 58.5 | 16.9 |

**GPAT builds creative assemblies.** To fully test the generalization abilities of GPAT, we provide unseen part shapes like a banana, hammers, and forks as parts to create novel targets such as a plane. Our model predicts creative assemblies given target shapes from unseen categories and non-exact parts, as seen in Fig. 8. The shapes are taken from PartNet [6], ModelNet40 [33], and YCB dataset [34].

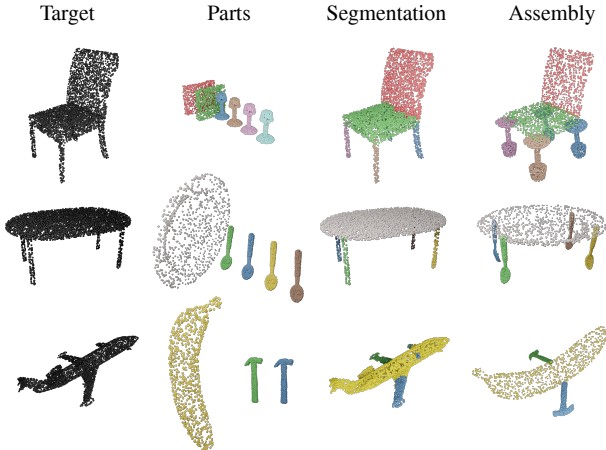

Figure 8: **Creative Assemblies.** Our model predicts creative assemblies given target shapes from unseen categories and non-exact parts. 1st row: a chair assembled with lamps as chair legs. 2nd row: a table assembled with a plate and spoons. 3rd row: a plane assembled with a banana and hammers.

**Oriented bounding boxes offer a sufficient pose estimator.** Once we obtain the target segments with GPAT, we compare with alternative methods to obtain the final poses and present the quantitative results in Tab. 4. The alternative methods include Go-ICP [30], DCP [35] (we take the released model pretrained on Modelnet40 [33]). Additionally, we adapt the previous work [3] by Li et al. to our task. Li ei al. consider targets represented as images, and train a model to produce 2D part segments and another GNN-based model to regress part poses. We produce 3D segments using our pretrained GPAT and train the GNN-based backbone proposed in DGL [4], an improved model compared to that used in [3]. We find the heuristics based on oriented bounding boxes to produce comparative or better results compared to more sophisticated alternative methods. Furthermore, we provide further analysis in the Appendix to show that the main bottleneck of the problem is segmentation as opposed to pose estimation.

| | Unseen Instance | | | | | | | | Unseen Category | | | | | | | |
| | Canonical Pose | | | | | | Random Pose | | | | | | Canonical Pose | | | | | | Random Pose | | | | | |
| | Precise Part | | | Imprecise Part | | | Precise Part | | | Imprecise Part | | | Precise Part | | | Imprecise Part | | | Precise Part | | | Imprecise Part | | |
| | CD | PA | SR | CD | PA | SR | CD | PA | SR | CD | PA | SR | CD | PA | SR | CD | PA | SR | CD | PA | SR | CD | PA | SR |
|---|---|---|---|---|---|---|---|---|---|---|---|---|---|---|---|---|---|---|---|---|---|---|---|---|
| GPAT-GoICP | **6.7** | **63.9** | **23.7** | 7.8 | 63.5 | 20.0 | **6.8** | **64.9** | **24.6** | 8.0 | 62.5 | 19.7 | **5.9** | **53.9** | 19.7 | **6.5** | 55.7 | 18.6 | **5.3** | **55.8** | 20.4 | **5.9** | 58.6 | 22.1 |
| GPAT-DCP | 18.2 | 37.3 | 6.2 | 18.6 | 32.1 | 2.1 | 18.7 | 36.0 | 5.5 | 18.4 | 32.0 | 2.8 | 15.4 | 33.1 | 3.4 | 16.7 | 30.1 | 1.0 | 15.3 | 33.4 | 4.4 | 16.9 | 30.5 | 2.3 |
| GPAT-DGL | 12.1 | 52.1 | 13.0 | 10.8 | 58.4 | 17.2 | 12.2 | 51.1 | 11.5 | 10.5 | 55.8 | 16.2 | 13.1 | 38.4 | 6.9 | 10.5 | 44.8 | 9.6 | 11.5 | 42.3 | 10.3 | 10.4 | 48.5 | 13.6 |
| GPAT-BB (Ours) | 7.6 | 61.6 | 23.2 | **7.2** | **64.8** | **26.0** | 7.8 | 60.8 | 21.7 | **7.8** | **64.3** | **26.0** | 7.1 | 53.4 | **20.1** | 6.6 | **56.3** | **21.7** | 7.6 | 52.2 | 18.8 | 6.9 | 55.6 | 19.8 |

Table 4: **Evaluation of the Pose Estimation Module.** We find the efficient heuristics based on oriented bounding boxes to produce comparative or better results compared to more sophisticated alternative methods.

**Segmentation Accuracy is the Main Bottleneck.** In Fig. 9, we plotted the mean success rate / part accuracy conditioned on the minimum segmentation accuracy. We find that when segmentation accuracy approaches perfect, average success rate and part accuracy approaches 90%, while the current overall numbers are around 20% and 60%, respectively. This shows that segmentation accuracy is still the main bottleneck of our method.

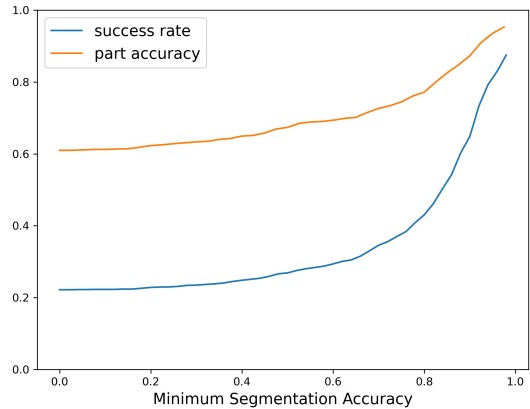

Figure 9: **Segmentation Accuracy is the Main Bottleneck.** Each point on the plot reads as "for all the data samples with minimum accuracy of x, the average success rate / part accuracy is y".

**Optimization is prone to local minima.** The optimization baseline (Opt) achieves the lowest chamfer distance (CD) in some scenarios, but its part accuracy (PA) and success rate (SR) are significantly lower. As seen in Fig. 10, directly optimizing the part poses to match the target often result in predictions at local minima where the predicted assembly matches the contour of the target, but the assembly makes no semantic sense.

**GPAT is applicable to part discovery.** GPAT is directly applicable to the task of part discovery, i.e., predict a part segmentation given a target [36], if we do not provide input parts. We show some qualitative results in Fig 11 to test GPAT's part discovery abilities. Given non-exact parts, PAT predicts accurate segmentations as usual. If we input identical blocks, which specifies the number of parts but provides little information about the part shapes, then GPAT predicts reasonable segmentations with the specified number of segments. Finally, we omit the input parts, and GPAT successfully discovers parts in the target shape.

**GPAT is aware of part scales.** Part assembly often involves parts that have the same geometry but different scales, so it is necessary for a model to discriminate parts of different scales to create correct assemblies. As an qualitative illustration in Fig. 12, we adjust the scale of the parts that have same geometry (the legs of chair/table), and the model correctly associates parts of different scales to the target to build the desired assemblies.

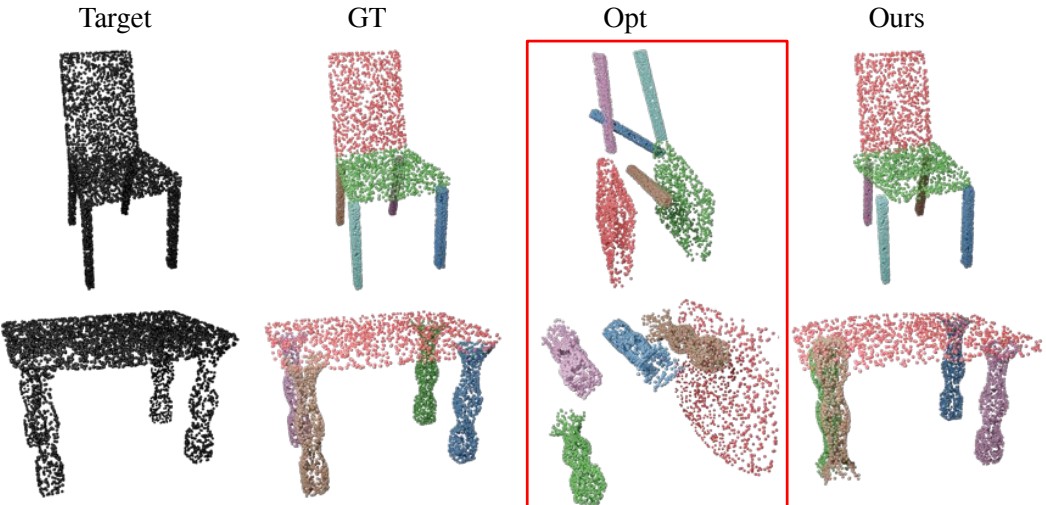

Figure 10: **Optimization is prone to local minima.**

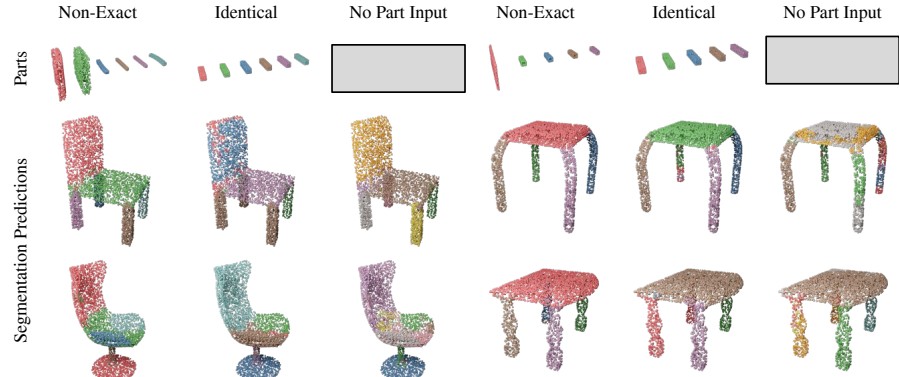

Figure 11: **Application to Part Discovery.** Given non-exact matching parts (Non-Exact), identical blocks (Identical), and no part point clouds input, GPAT predicts reasonable part segmentations of the target.

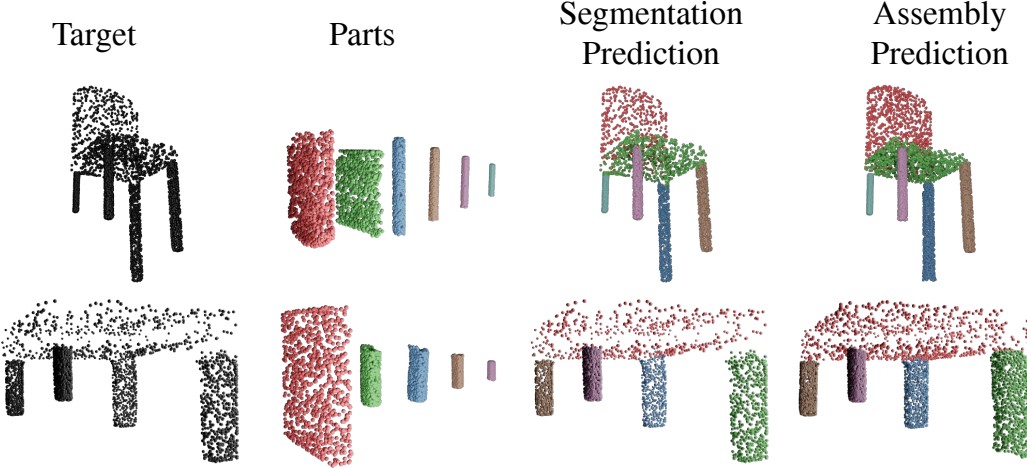

Figure 12: **Sensitivity to Scale**. The legs of the chair/table are manually scaled, and the model correctly associate parts of the same shape but different sizes.

## 8.2   Data and Training Details

We use Furthest Point Sampling (FPS) to sample 1,000 points for each part point cloud and 5,000 points for each target point cloud. Following the previous work [37], we also zero-center all the point clouds, and align the principle axes of the part point clouds with the world axes using Principle Component Analysis (PCA). Additionally, we similarly use axis-aligned bounding boxes to obtain 3-dimensional sizes of the part, and two parts are considered geometrically equivalent if they have the same part type as labeled by the PartNet dataset [38] and same sizes up to a small threshold.

In our training, we down-sample the target point features by a factor of 10, so for each sample, we obtain 500 target point features. We use a feature dimension of 256 and we use 8 GPAT layers, with $k$ values of $16, 16, 32, 32, 64, 64, 500, 500$. We use Adam [39] with a learning rate of 0.00004, a batch size of 36. We train for 2000 epochs in total.

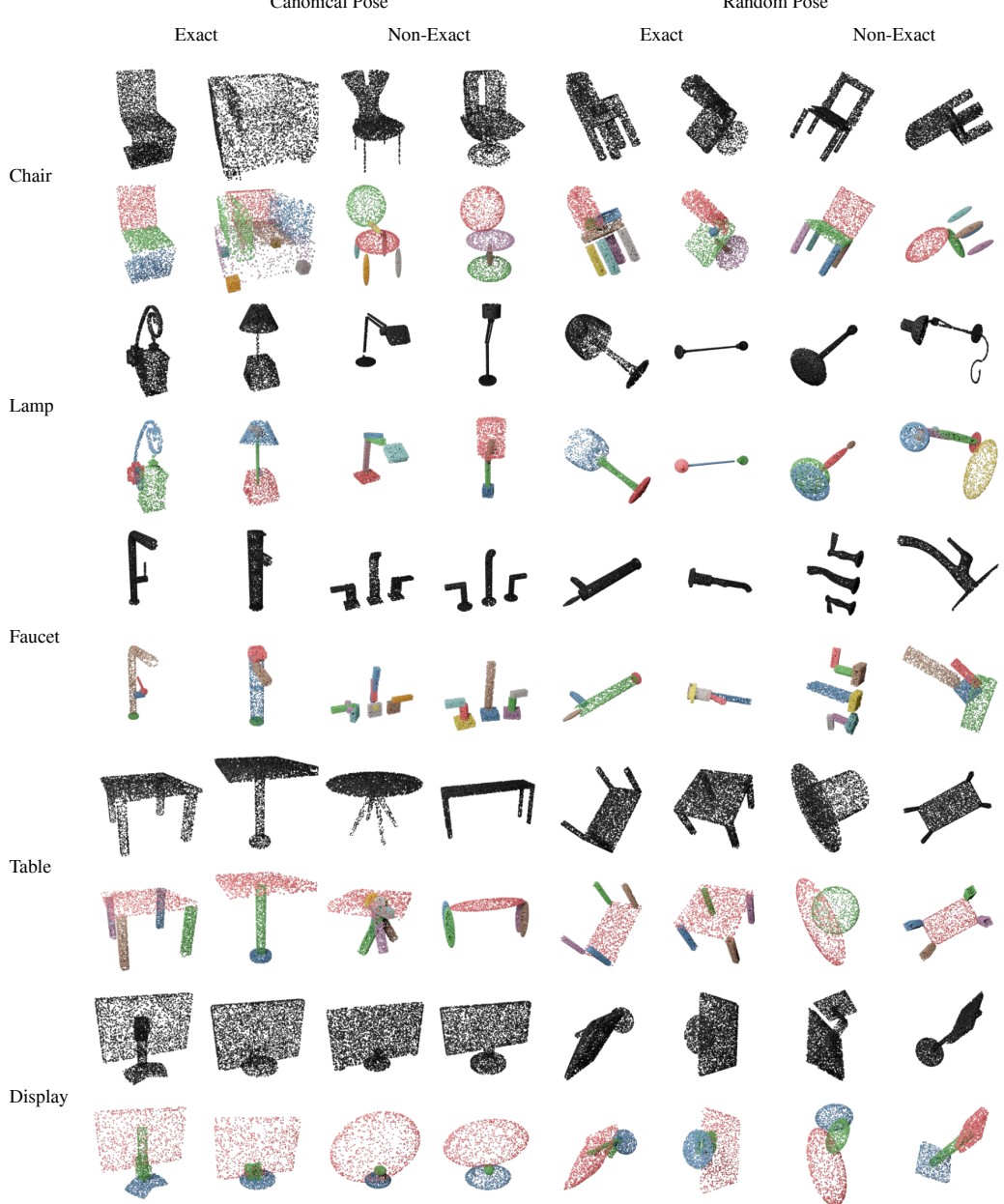

Figure 13: **Qualitative Results**. Chairs, lamps, and faucets are seen during the training. Tables and displays are unseen categories. The first row of each category displays targets in black, and the second row shows our predictions.

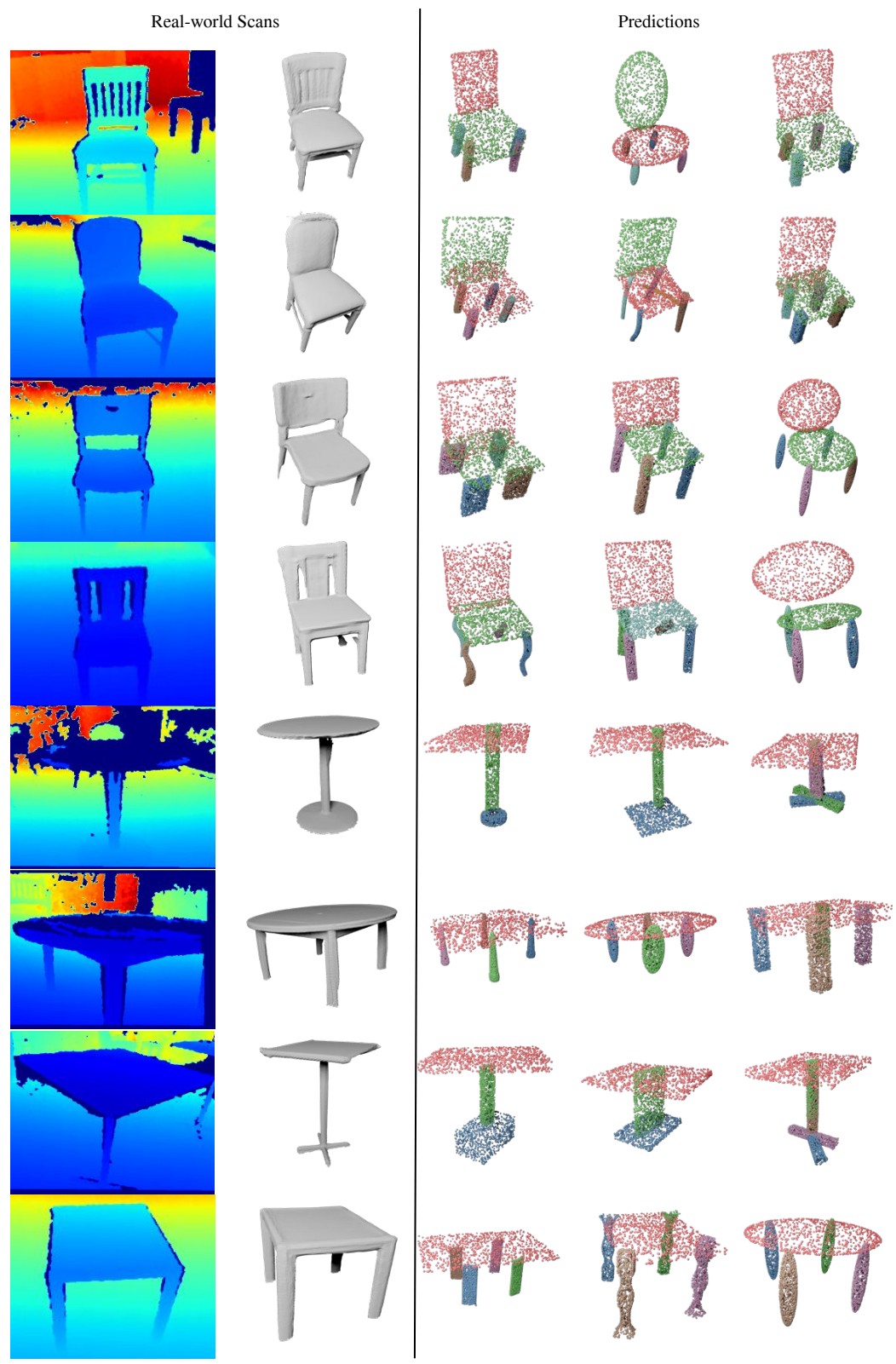

Figure 14: **Results on Real-world Data**. Non-exact parts from the same category as the target point cloud, which are teal-world scans taken from Redwood dataset [32].

