# OpenReview forum: "Rearrangement Planning for General Part Assembly"
_robot-learning.org/CoRL/2023/Conference — CoRL 2023 Oral_

### Official Review · Reviewer_JQms · 2023-07-16

**Confidence:** 4
**Originality:** Very Good
**Technical Quality:** Very Good
**Clarity Of Presentation:** Excellent
**Impact:** 4

**Recommendation:**

Strong Accept: I recommend accepting the paper and will argue for my recommendation even if other reviewers hold a different opinion.

**Review:**

High quality paper, clearly written, original, and providing a significant step forward on the part assembly problem.

The evaluation is a bit limited, as the method was trained on "chairs, lamps, and faucets" and tested on "tables and displays". Is there a reason why only so few categories were used? What would happen if one swaps some categories between train and test sets (e.g., swap tables and chairs)?

**Quality Of The Limitations Section:**

Limitations are addressed clearly

**Questions For Rebuttal:**

The paper argues that RL "has success in building part assembly for **fixed targets** [...] or **seen categories** [...]" (lines 64--65), whereas this paper allows for generalization to "**novel target shapes** without additional annotation". However, I would like to point out that there was also a demonstration of RL+GNN method which can assemble various shapes even with a robot arm. Namely, citing from the abstract of [1]:

"Combining structured representations with model-free RL and Monte-Carlo planning **allows agents to operate with various target shapes** and building block types. We design a hierarchical control framework that learns to sequence the building blocks to **construct arbitrary 3D designs** and ensures their feasibility, as we plan the geometric execution with the robot-in-the-loop."

Of course, the method proposed in the current submission is quite different, as it works on point clouds and on more semantically grounded shapes which have parts, whereas the referenced paper builds arbitrary shapes out of given blocks. Nevertheless, both approaches tackle the part assembly problem, and I would recommend to slightly rephrase the introduction and related work section on Specialized Part Assembly to acknowledge that other methods also attempted to generalize to novel shapes, and emphasize how the proposed method is different.

[1] Funk, N., Chalvatzaki, G., Belousov, B., & Peters, J. (2022). Learn2assemble with structured representations and search for robotic architectural construction. In Conference on Robot Learning (pp. 1401-1411). PMLR.

**Robotics Focus:**

Highly relevant to robotics but no hardware experiments

**Summary Of Paper:**

The paper formulates the general part assembly problem as a goal-conditioned shape rearrangement task, and proposes a Transformer-based architecture that processes input point clouds in a fine-to-coarse manner in two steps: predict a segmentation of the target shape and subsequently infer the pose of each part. Experiments on PartNet demonstrate the method and show its superior performance compared to baselines. Additionally, generalization to real-world scans is shown on the redwood dataset.

**Summary Of Recommendation:**

A strong high-quality paper that proposes a general approach for solving part assembly problems on point clouds. In spite of the demonstrations being presented on a restricted set of categories, the results appear convincing and promising. Combined with task and motion planning, this method can contribute to a general autonomous robotic assembly system.

---

### Official Review · Reviewer_wPeg · 2023-07-17

**Confidence:** 4
**Originality:** Good
**Technical Quality:** Very Good
**Clarity Of Presentation:** Excellent
**Impact:** 3

**Recommendation:**

Weak Accept: I recommend accepting the paper, but will not argue for my recommendation if the majority of other reviewers have a different opinion.

**Review:**

Strengths:
- The paper presents a novel and interesting task of general part assembly.
- The proposed method is technically novel, specifically in proposing to do segmentation first and using optimization-based pose estimations for better generalization capabilities.
- Experiments show clear advantages over baseline methods both qualitatively and quantitatively.

Weaknesses:
- It's a bit unclear why the proposed method would not overfit the training categories as the baselines do, since the PointNet/transformer backbones may also be subject to overfitting as the baseline methods. Or in other words, may the baseline methods also be able to generalize well to novel categories with increased numbers of the training categories? Can the authors show results on more test categories that are dissimilar to the training categories?
- The proposed method is similar to [3] which also proposed to do segmentation first but it's in 2D. It would be better to add a comparison to an adapted version of [3] to segment the input shapes in 3D.

**Quality Of The Limitations Section:**

Limitations are addressed clearly

**Questions For Rebuttal:**

Besides the above points in weaknesses, I have two clarification questions:
- in Fig. 6, the second row, why the back slats are not fitted?
- in Fig. 2, if the two "wing" parts are the same geometry, how to learn different features for them so that the green one corresponds to the left wing, and the blue one corresponds to the right wing, instead of the other way around?

**Robotics Focus:**

Relevant but unlikely to deploy to hardware in near future

**Summary Of Paper:**

This paper presents a novel method for tackling the task of 3D shape part assembly planning that can generalize across category boundaries. Given as inputs a 3D target shape point cloud and a set of 3D part shape point clouds, the system outputs the part segmentation and then the part poses of the input 3D parts to assemble the target shape. The parts can be the exact parts or non-exact parts (e.g., using 4 spoons and a cuboid to assemble a table shape). The core idea is to use PointNet to extract per-point features in the input shape and per-part features among the input parts and then employ a multi-attention transformer across the shape points and parts first to perform part segmentation and then optimize for the output part poses. The authors compared the proposed method to several baseline methods using the PartNet parts and show better and more generalizable results than baselines. Results on real scans are also presented. Failure cases and limitations are well discussed.

**Summary Of Recommendation:**

Overall, this is a good paper proposing a new interesting and important problem and presenting a technically novel method with good experimental results supporting its contributions. I'd like to see my questions addressed in the rebuttal, but they are minor points only serving as suggestions on potentially improving the paper.

Post rebuttal: Thanks for authors' rebuttal. I keep my score as Weak Accept.

---

> ### Author Response · Authors · 2023-08-11
> **Response 2/2**
>
> >  How to learn different features for them so that the green one corresponds to the left wing, and the blue one corresponds to the right wing, instead of the other way around?
>
> This happened to be a happy coincidence :) In fact, the pointnet backbone does not learn different features for the geometrically equivalent parts and different initialization and randomization can lead to entirely different results (i.e., the wings switch place). In Fig. 3, we can see that the predicted and the ground-truth poses are often different for the equivalent parts, for example, the armrests in the 1st column and the table legs in the 7th column.
> This keen observation of the reviewer highlights the inherent combinatorial complexity and multi-modality of the task. Due to this challenge, we proposed to use MultiScaleAttention to predict consistent part segmentations. As seen in Fig. 5, without the MultiScaleAttention module, the model fails to handle the equivalent parts correctly.
>
> [1] Thomas, Nathaniel, et al. "Tensor field networks: Rotation-and translation-equivariant neural networks for 3d point clouds." arXiv preprint arXiv:1802.08219 (2018).
>
> [2] Chen, Haiwei, et al. "Equivariant point network for 3d point cloud analysis." Proceedings of the IEEE/CVF conference on computer vision and pattern recognition. 2021.
>
> [3] Zhu, Minghan, Maani Ghaffari, and Huei Peng. "Correspondence-free point cloud registration with SO (3)-equivariant implicit shape representations." Conference on Robot Learning. PMLR, 2022.
>
> [4] Y. Li, K. Mo, L. Shao, M. Sung, and L. Guibas. Learning 3d part assembly from a single image. In European Conference on Computer Vision, pages 664–682. Springer, 2020.
>
> [5] G. Zhan, Q. Fan, K. Mo, L. Shao, B. Chen, L. J. Guibas, H. Dong, et al. Generative 3d part assembly via dynamic graph learning. Advances in Neural Information Processing Systems, 33:6315–6326, 2020.

---

> ### Author Response · Authors · 2023-08-14
> **Follow-up on the Rebuttal**
>
> Dear Reviewer,
>
> We’d like to reach out again to check if there are additional questions or concerns about our rebuttal that we can address before the reviewer-author discussion period ends on August 15. Thanks again for taking the time to read our work and provide helpful feedback!
>
> Paper Authors

---

### Official Review · Reviewer_v4vx · 2023-07-18

**Confidence:** 4
**Originality:** Fair
**Technical Quality:** Good
**Clarity Of Presentation:** Very Good
**Impact:** 3

**Recommendation:**

Weak Accept: I recommend accepting the paper, but will not argue for my recommendation if the majority of other reviewers have a different opinion.

**Review:**

Pros:
+ The paper is clearly written and easy to understand
+ The problem is interesting and important
+ The authors present many ablations, although these could be improved (see below)
+ The proposed solution appears well executed

Cons:
- (MINOR) The authors call this an assembly problem, but in reality it is closer to object decomposition.  The model only predicts a single point-cloud to part assignment, without any step-by-step analysis for how to construct the object, assuming that the parts may be connected in any arbitrary order by simply moving them to the correct 3D pose.  Similarly Figure1 labels the model's estimate an "Assembly Plan." The word "plan" suggests a sequential set of actions, which is not considered here.  While similar problems in prior work have also labelled themselves "assembly" problems, it would be beneficial to make this distinction clear by instead using terms like "decomposition" and "estimation" rather than "assembly" and "plan."
- (MODERATE) The authors' method consists of two steps: first compute an assignment from point cloud to the provided parts, then use a very simple bounding-box alignment mechanism to estimate the pose of each part.  With this in mind, the ablations would be better organized if these two components were isolated and handled independently.
- (IMPORTANT) Limited originality.  The high level problem and solution concept is quite similar to Li 2020 (citation [3]), but uses point-clouds instead of an image, with previously unseen categories.  The model components are also fairly standard which does not make a strong case for the practical utility of the paper.
- (IMPORTANT) While this might be used as a component of some robotics system, the connection to robotics in this paper is somewhat limited.  No experiments with real or simulated robots are presented.  However, I am not sure that robotics experiments would improve the quality of the paper, since the focus seems closer to 3D vision.  A vision conference may be a better venue for this work.

**Quality Of The Limitations Section:**

Limitations are addressed clearly

**Questions For Rebuttal:**

1. How could the system be adapted to problems requiring sequential planning?
2. How would this be incorporated into a larger robotic system?

**Robotics Focus:**

Relevant but unlikely to deploy to hardware in near future

**Summary Of Paper:**

The authors propose a new task consisting of building a previously unseen target objects using a list of new parts specified as point clouds.  In the authors' proposed problem, the parts may not exactly fit the target model, so the network must learn to assemble something as close as possible to the target shape.  The authors also propose a transformer-based model to address this task using multi-scale attention and cross-attention to part representations.

**Summary Of Recommendation:**

The paper is well written and executed, but the limited originality and lack of robotics focus weigh it down.  I do not feel strongly about rejecting this paper though, and would be willing to adjust my score upwards if the authors or more importantly the other reviewers make a strong case that these limitations are not important to the CoRL audience.

Post Rebuttal: I am bumping this up to weak-accept.  I appear to have the lowest review of this paper, and my complaints were about the originality and the lack of robotics focus.  I wasn't sure if this was the right way to evaluate the paper, and these issues didn't seem to have been a problem for the other reviewers, so I am adjusting upward.

---

> ### Author Response · Authors · 2023-08-11
> **Response 2/2**
>
> > How could the system be adapted to problems requiring sequential planning? How would this be incorporated into a larger robotic system?
>
> The concern is definitely valid, and our motivation at the beginning of the project was to build a generalist robot system for part assembly given some arbitrary targets and parts in the wild. However, we soon realized that the primary bottleneck to achieve this goal is the initial perception and planning module, where considerable prior work in the area focuses on either 1) given target object poses or 2) known object parts and target(s). Therefore in this work, we decided to focus on a deep dive of this subproblem to help relax some of the assumptions made in prior work. Given the desired part poses inferred by our algorithm, it may then be possible to build a modular system for part assembly by leveraging prior work on assembly sequencing and path planning [4, 5, 6], which can be based on a simple sampling-based planner, or reinforcement learning. Rather than searching over an immeasurable action space and state space, the agent can start with knowledge of the predicted goal poses of each part.
>
> [1] J. Yang, H. Li, D. Campbell, and Y. Jia. Go-icp: A globally optimal solution to 3d icp point-set registration. IEEE transactions on pattern analysis and machine intelligence, 38(11): 2241–2254, 2015.
>
> [2] . Wang and J. M. Solomon. Deep closest point: Learning representations for point cloud registration. In Proceedings of the IEEE/CVF international conference on computer vision, pages 3523–3532, 2019.
>
> [3] Y. Li, K. Mo, L. Shao, M. Sung, and L. Guibas. Learning 3d part assembly from a single image. In European Conference on Computer Vision, pages 664–682. Springer, 2020.
>
> [4] Mohd Fadzil Faisae Rashid, Windo Hutabarat, and Ashutosh Tiwari. 2012. A review on assembly sequence planning and assembly line balancing optimisation using soft computing approaches. The International Journal of Advanced Manufacturing Technology 59, 1 (2012), 335–349.
>
> [5] Y. Tian, J. Xu, Y. Li, J. Luo, S. Sueda, H. Li, K. D. Willis, and W. Matusik. Assemble them all: Physics-based planning for generalizable assembly by disassembly. ACM Transactions on Graphics (TOG), 41(6):1–11, 2022.
>
> [6] Ellips Masehian and Somayé Ghandi. 2021. Assembly sequence and path planning for monotone and nonmonotone assemblies with rigid and flexible parts. Robotics and Computer-Integrated Manufacturing 72 (2021), 102180.
>
> [7] Batra, Dhruv, et al. "Rearrangement: A challenge for embodied ai." arXiv preprint arXiv:2011.01975 (2020).

---

> ### Author Response · Authors · 2023-08-14
> **Follow-up on the Rebuttal**
>
> Dear Reviewer,
>
> We’d like to reach out again to check if there are additional questions or concerns about our rebuttal that we can address before the reviewer-author discussion period ends on August 15. Thanks again for taking the time to read our work and provide helpful feedback!
>
> Paper Authors

---

> > ### Comment · Reviewer_v4vx · 2023-08-14
> > **Thanks**
> >
> > Hey, thanks for the detailed feedback, this all makes sense.  I will incorporate this info into my updated review.

---

### Official Review · Reviewer_sArU · 2023-07-19

**Confidence:** 4
**Originality:** Good
**Technical Quality:** Very Good
**Clarity Of Presentation:** Good
**Impact:** 3

**Recommendation:**

Weak Accept: I recommend accepting the paper, but will not argue for my recommendation if the majority of other reviewers have a different opinion.

**Review:**

Strengths:
1. The paper is well-written and easy to follow.
2. The paper introduces a transformed-based framework to tackle the part assembly task with convincing results both qualitatively and quantitively.

Weakness:
1. The title of the paper “general part assembly PLANNING” is misleading. There is no classical planning incorporated in this manuscript.
2.  It’s not clear to the reviewer how effective the proposed pose estimation using the oriented-bounding boxes. It would strengthen the paper if more analysis and comparison is provided.


**Quality Of The Limitations Section:**

Additional details required

**Questions For Rebuttal:**

Q1. See weakness 2.

Q2. Section 4. Line 184. “Due to possible geometric equivalence between parts (e.g., the legs of a chair), we enumerate all possible labels of geometrically equivalent parts to obtain different GT poses and take the highest accuracy value.” Does the dataset of Part provide geometrically equivalent parts annotation? If not, how do the authors gather such annotations?

Q3. In Section 5, Line 242. “We use the real-world scans from redwood  dataset [29] as targets and part point clouds from PartNet.” What is the performance if both the target point cloud and part point clouds are scanned from the real world?



**Robotics Focus:**

Relevant but unlikely to deploy to hardware in near future

**Summary Of Paper:**

This manuscript introduces a transformed-based neural network model for the general part assembly task. The model takes as inputs a complete target point cloud and the set of parts to be assembled,  and outputs the segmentation of the target point cloud and associated parts for each segmented region. Then a pose estimation module is utilized to generate the part pose. The authors also conduct extensive experiments to verify the effectiveness of the proposed pipeline and compare it with other methods.

**Summary Of Recommendation:**

This paper proposes a transformed-based neural network to tackle the general part assembly task by formulating it as a goal-conditioned shape-matching problem, which is kind of novel. However, the topic is more related to computer vision instead of robotics.

---

> ### Author Response · Authors · 2023-08-14
> **Follow-up on the Rebuttal**
>
> Dear Reviewer,
>
> We’d like to reach out again to check if there are additional questions or concerns about our rebuttal that we can address before the reviewer-author discussion period ends on August 15. Thanks again for taking the time to read our work and provide helpful feedback!
>
> Paper Authors

---

### Author Response · Authors · 2023-08-11
**Response summary**

We would like to thank the reviewers for their time and effort in helping us improve our work. We are glad that the reviewers acknowledge the value of the work, and we made relevant revisions and included more experiments and analysis as suggested by the reviewers. We uploaded the revised manuscript with individual responses, and below is a summary of our response and major updates.

**Evaluation of the Pose Estimation Module**

For the new revision, we have included additional comparisons to two alternative pose estimation methods, Go-ICP and DCP (Table 2 and Sec. 5). Surprisingly, we find that the simple heuristics based on oriented bounding boxes continue to produce comparable (or at times better) results compared to the more complex alternatives. We have also included additional empirical observations in the appendix (Fig. 9) that interestingly suggest that the success rate of the overall framework appears to be predominantly influenced (and bounded) by the accuracy of the target segmentation step, as opposed to the pose estimation step.

**Comparison with the Previous Work by Li et al.**

We adapt the previous work to predict part poses and include the evaluation results in Table 2. We found that the adapted version works significantly better than directly regressing poses using a GNN backbone, but still performs worse compared to a simple pose estimation method.

**Addressed the Confusion of the Term “Planning”**

We thought it would be worthwhile to keep "planning" but rephrase it to "rearrangement planning" – which aligns with established definitions previously proposed by the Task and Motion Planning (TAMP) community ("Rearrangement: A Challenge for Embodied AI, "Batra et al., 2020).

**Connection to Robotics**

Our motivation at the beginning of the project was to build a generalist robot system for part assembly given some arbitrary targets and parts in the wild. However, we soon realized that the primary bottleneck to achieve this goal is the initial perception and planning module, where considerable prior work in the area focuses on either 1) given target object poses or 2) known object parts and target(s). Therefore in this work, we decided to focus on a deep dive of this subproblem to help relax some of the assumptions made in prior work. Given the desired part poses inferred by our algorithm, it may then be possible to build a modular system for part assembly by leveraging prior work on assembly sequencing and path planning, which can be based on a simple sampling-based planner, or reinforcement learning.

---

### Decision · Program_Chairs · 2023-08-30

**Decision:**

Accept (Oral)

**Comment:**

The paper introduces a novel approach to the problem of 3D shape part assembly planning by formulating it as a goal-conditioned shape rearrangement task and proposing a Transformer-based architecture. The proposed method shows promise in solving this task, and it has been received positively by reviewers, with both strengths and areas for improvement highlighted.
Strengths identified by reviewers include the quality of the paper, its clarity, and the originality of the approach. The method's technical innovation and potential for impact in the field have been highlighted. The paper also addresses limitations clearly and provides solid experimental results.
Reviewers have raised questions and suggestions regarding the evaluation setup. Specifically, concerns were raised about the limited number of categories used for testing and the potential effects of swapping training and testing categories. The authors have responded by explaining the limitations of available dataset categories and conducting a preliminary experiment to evaluate the impact of category choices.
Another point of discussion is the similarity of the proposed method to prior work in terms of generalizing to novel shapes. A related work by Funk et al. [1] was brought to the authors' attention, demonstrating the importance of acknowledging such related works and highlighting the differences between them.
The paper is considered to be of high quality, presenting a significant step forward in addressing the part assembly problem with promising results and innovation.
Post-rebuttal comments show that reviewers have maintained their recommendations, indicating that the authors' responses effectively addressed their concerns and reinforced the paper's value.